# An Edge Server Placement Method Based on Reinforcement Learning

**DOI:** 10.3390/e24030317

**Published:** 2022-02-23

**Authors:** Fei Luo, Shuai Zheng, Weichao Ding, Joel Fuentes, Yong Li

**Affiliations:** 1School of Information Science and Engineering, East China University of Science and Technology, Shanghai 200237, China; luof@ecust.edu.cn (F.L.); jesse2017315@gmail.com (S.Z.); 2Department of Computer Science and Information Technologies, Universidad del Bío-Bío, Chillán 3780000, Chile; jfuentes@ubiobio.cl

**Keywords:** edge computing, markov decision process, reinforcement learning, access delay, workload balance

## Abstract

In mobile edge computing systems, the edge server placement problem is mainly tackled as a multi-objective optimization problem and solved with mixed integer programming, heuristic or meta-heuristic algorithms, etc. These methods, however, have profound defect implications such as poor scalability, local optimal solutions, and parameter tuning difficulties. To overcome these defects, we propose a novel edge server placement algorithm based on deep q-network and reinforcement learning, dubbed DQN-ESPA, which can achieve optimal placements without relying on previous placement experience. In DQN-ESPA, the edge server placement problem is modeled as a Markov decision process, which is formalized with the state space, action space and reward function, and it is subsequently solved using a reinforcement learning algorithm. Experimental results using real datasets from Shanghai Telecom show that DQN-ESPA outperforms state-of-the-art algorithms such as simulated annealing placement algorithm (SAPA), Top-K placement algorithm (TKPA), K-Means placement algorithm (KMPA), and random placement algorithm (RPA). In particular, with a comprehensive consideration of access delay and workload balance, DQN-ESPA achieves up to 13.40% and 15.54% better placement performance for 100 and 300 edge servers respectively.

## 1. Introduction

Mobile cloud computing is the combination of cloud computing and mobile computing to bring rich computational resources to end mobile users, network operators, and cloud computing providers. It has also become one of the new hotspots of the mobile Internet services, where problems such as network overload/underload uplink and downlink arise between the cloud and the mobile devices [1]. Mobile edge computing (MEC) addresses these problems by localizing communication, storage and computing resources near the user side, and providing resources with lower network latency and reduced network congestion.

MEC and important parts of its related research have been focused on performance improvements for task offload [2,3,4], service migration [5,6,7], and quality-of-service (QoS) prediction [8]. Most of the published work consider the premise that the location of edge server has been previously determined, without considering edge server placement. However, improper edge server placement may cause some edge servers to be overloaded or underloaded [9,10,11], which leads to a QoS decrease. In the contrast, a reasonable edge server placement scheme can provide edge servers more balanced workloads, and can also make the overall delay lower, achieving better QoS for users [12,13].

Currently, the edge server placement problem (ESPP) is mainly approached by transforming it into a multi-objective optimization problem, which is further solved by mixed integer programming, heuristic algorithms or meta-heuristic algorithms [11,12,14]. However, these methods have profound limitations and performance implications. For instance, mixed integer programming cannot solve large-scale problems in a limited time [13]. Heuristic algorithms depend on their specific problem models, and it is easy to fall into local optimal solutions. Meta-heuristic algorithms are not easily reusable due to a significant amount of problem-specific parameter tuning [15].

Different from the above methods, this paper proposes a novel edge server placement algorithm based on deep reinforcement learning, dubbed DQN-ESPA. First, the ESPP is modeled as a Markov decision process (MDP), where the goal is to balance edge server workloads and minimize the access delay between the mobile user and edge server. Then, reinforcement learning is applied to find the optimal solution. Specifically, an agent continuously learns online and improves its placement decisions through a trial-and-error learning scheme. The introduction of a deep neural network (DNN) helps to deal with the exponential growth of states and actions when exploring optimal control in high-dimensional spaces, known as the curse of dimensionality [16].

The main contributions of this paper can be summarized as follow:A new solution model is proposed for the ESPP based on MDP and reinforcement learning. In this model, location sequences of edge servers are modeled as states, decisions of move direction of edge servers are modeled as actions, and the negative access delay and standard deviation of workloads are modeled as rewards. By maximizing the cumulative long-term reward, the ESPP can be solved by the reinforcement learning algorithm.An edge server placement algorithm based on deep q-network (DQN), named DQN-ESPA, is proposed to solve the ESPP by combining a deep neural network and a Q-learning algorithm.

The remainder of this paper is organized as follows. Related work is presented in Section 2. The ESPP is modeled in Section 3, which abstracts the edge computing architecture into a network structure composed of users, base stations and edge servers. The details of DQN-ESPA algorithm are described in Section 4. In Section 5, experiments are carried out and the performance of the proposed algorithm is evaluated with the other state-of-date algorithms. Finally, conclusions are drawn and the directions of future work are presented in Section 6.

## 2. Related Work

To meet the requirements of high resource utilization and small network delay, the ESPA selects an appropriate geographical location for the edge server in MEC. For convenience and consistency, edge devices with rich computing and storage resources, such as the roadside units (RSUs), cloudlets and the MEC edge servers, are uniformly called as edge servers [17].

Classical approaches have been used for the ESPP, such as the K-Means placement algorithm (KMPA) [18], the Top-K placement algorithm (TKPA) and the Random placement algorithm (RPA) [12,19]. Taking the network coverage and deployment cost as optimization indexes, Liang et al. [20] studied the deployment of RSUs in a two-dimensional vehicle network using linear programming. Wang et al. [21] introduced the concept of centrality in social networks into the deployment of RSUs, and proposed a method of RSUs deployment based on centrality. First, the RSUs deployment problem was abstracted into a linear programming problem, and then the RSUs problem was transformed into a 0-1 knapsack problem. Compared to the random deployment method, the efficiency of RSUs deployment was significantly improved. At present, a large number of research activities on RSUs placement are mainly from the perspective of communications [20,22,23,24], while less attention is paid to the demand of computing resources for vehicle applications. Premsankar et al. [25] considered the placement of RSUs from two aspects of network coverage and computing resources, and solved the placement problem of RSUs using mixed integer programming. Zhang et al. [14] put forward an asynchronous particle swarm optimization algorithm to optimize the RSUs deployment from the perspective of positioning accuracy and deployment cost, hoping to use the least RSUs to obtain the best vehicle positioning accuracy.

Some approaches on the optimal configuration of cloudlets were also studied. Ma et al. studied in [9] the problem of cloudlet placement in large-scale wireless metropolitan area networks. By abstracting the problem of cloudlet placement into the problem of integer linear programming, and considering the limitation that integer linear programming cannot solve the large-scale problem, the algorithm of k-medoids was used to solve the problem of cloudlets’ placement. Xu et al. [10] considered the placement of cloudlets with different computing resources, and proposed a fast and effective heuristic algorithm to minimize access delay. Authors in [11] considered the impact of user’s dynamic request on cloudlet placement, adopting approximate and iterative algorithms to obtain the optimal placement scheme. Observing that the deployment cost was not considered in some studies, Fan et al. [26] established a linear programming model by considering both the deployment cost and the access delay.

In view of the strategies and configuration used in the problems listed above, subsequent research focused on workload balance [27] or access delay [28,29,30]. Zhao et al. [12] placed edge servers comprehensively considering both the workload balance and the access delay. By constructing the placement as a multi-objective optimization problem, they solved the placement of edge servers through mixed integer planning. Subsequently, Guo et al. [13] proposed an approximation algorithm based on K-means and mixed integer quadratic programming. From the perspective of minimizing the number of edge servers deployed and QoS requirements, Zeng et al. [31] utilized the simulated annealing algorithm and the greedy placement algorithm, named as SAPA.

Considering the ESPP in particular, the described work have either used mixed integer programming algorithms, heuristic algorithms, and/or meta-heuristic algorithms. However, different problems arise for various scenarios commonly found in MEC, i.e. mixed integer programming algorithms cannot solve large-scale problems in a limited time; heuristic algorithms rely on specific problem models and easily to fall into the local optimal solution; meta-heuristic algorithms require high levels of parameter-tuning.

Reinforcement learning is commonly used to solve sequence decision-making problems and find the optimal strategy by maximizing long-term rewards [32]. In edge computing, it has been used to solve task offloading and resource allocation problems [4,33,34,35]. To the best of our knowledge, there are no studies on edge server placement in MEC environments using reinforcement learning. Therefore, this paper proposes a model and algorithm to solve the ESPP using reinforcement learning, providing users with lower latency and more balanced workload among edge servers. Typical comparison algorithms are used to demonstrate the superiority of the proposed algorithm in sequential decision-making circumstances, such as SAPA, KPMA, TKPA and RPA. Especially, as a variant of top-k algorithms, the bubble-sort algorithm is implemented in the TKPA experiments for comparison.

## 3. System Model

In this section we introduce the MEC model, which simplifies the edge computing architecture into a network structure consisting of users, base stations and edge servers. Then, a detailed description of the ESPP is provided. Additionally, we formalize the workload balance model and access delay, which will be used to evaluate the performance of the proposed algorithm.

### 3.1. MEC Model

In order to reduce the delay and solve the user’s limited processing capacity, MEC migrates the computing and storage capability from the remote data center to the network edge close to the user. In the mobile edge computing environment, the edge computing system model can be defined as an undirected graph G=(V,E), which contains users, base stations and edge servers. As it is shown in Figure 1, $bs_{i}$ represents a base station, $es_{i}$ represents an edge server and $md_{i}$ represents a mobile device. The mobile device forwards the request to the edge server through the base station, and the edge server returns the processed result to the mobile device through the base station. The set of points in the undirected graph *G* is represented by *V*. V=B∪S, where *B* is the set of base stations and *S* is the set of edge servers placed in candidate locations. *E* is the set of connected edges between the base station and the edge server in *G*. User requests are forwarded to the appropriate edge server nearby through the base station, and the edge server returns the processed results to the user.

### 3.2. Problem Description

In an edge computing system, all edge servers and base stations are represented by (S,B), where $S=\{s_{1},s_{2},\ldots ,s_{m}\}$ represents a set of *m* edge servers. $B=\{b_{1},b_{2},\ldots ,b_{n}\}$ represents a set of *n* base stations. The workload of each base station $b_{i}$ is represented as $w_{i}$, and the distance between the base station $b_{i}$ and the edge server $s_{j}$ is represented as $d(b_{i},s_{j})$. In order to simplify the problem, five assumptions are given as follows.
Each edge server is homogeneous and has the same processing and storage capabilities.Let the set of base stations covered by each edge server be $B_{i}$, the set of base stations covered by each edge server does not intersect, i.e., $B_{i}\cap B_{j}=Ø$. The union of the set of base stations covered by all edge servers is $B=\sum_{i=1}^{m}B_{i}$.An edge server is placed next to a base station, and *n* base stations have *n* positions.Only one edge server can be placed in each location.Each edge server is only responsible for processing requests uploaded by base stations within the signal coverage, and each base station within the signal coverage of each edge server can only forward requests to the edge server.

*m* edge servers are placed in the location set L with *m* locations through the placement strategy, where L=$\left\{l_{1},l_{2},\ldots ,l_{m}\right\}$. Therein, $l_{i}$ represents the placement location of the edge server $s_{i}$, which minimizes the delay and workload balance of the entire system. The workload balance and access delay of the system are described below.


*Workload balance*
The edge server $s_{i}$ is responsible for processing network service requests forwarded through all base stations in the base station set $B_{i}$. The workload of each base station is represented as $w_{j}$, and the workload of the edge server $s_{i}$ is represented as $W_{i}$ which is the sum of the workloads of all base stations in $B_{i}$, as shown in Formula (1).
(1)$$W_{i}=\sum_{b\in B_{i}}w_{j}$$Then the standard deviation of all the edge servers’ workloads is shown in Formula (2), where $\bar{W}=\frac{1}{m}\sum_{i=1}^{m}W_{i}$ is the average workload of m edge servers.
(2)$$WSD=\sqrt{\frac{\sum_{i=1}^{m}{\left(W_{i}-\bar{W}\right)}^{2}}{m}}$$
*Access delay*
Each base station can directly request the edge server to obtain network services through the link connection. Assuming that the positions of the base station and the edge server are denoted as $l_{b}$ and $l_{s}$, respectively, the length of the connecting edge between the two can be obtained based on the longitude and latitude information of the base station, as shown in Formula (3). In this paper, we use the length of the edge to measure the data transmission delay between the base station and the edge server.
(3)$$d\left(b_{i},s_{j}\right)={∥l_{b_{i}}-l_{s_{j}}∥}_{2}$$Then the average access delay is represented as Formula (4).
(4)$$MD=\frac{\sum_{i=1}^{m}\sum_{b_{j}\in B_{i}}d\left(b_{j},s_{i}\right)}{n}$$

## 4. Algorithm Design

Reinforcement learning consists of an agent and an environment. The agent can change the current environmental state by executing an action, and the environment gives the agent a reward according to the action executed by the agent. The goal of reinforcement learning is to learn the optimal strategy by maximizing the cumulative long-term reward, which is usually established as an Markov decision process model. The MDP model is denoted as a state-action-reward-state transition sequence, specifically a 5-tuple (S,A,T,R,γ) described as follows:*S* is a limited state space.*A* is a limited action space.T:S∗A∗S→[0,1], it is the state transition model. Specifically, $T(s,a,s^{\prime })$ represents the probability distribution that the agent transitions to state $s^{\prime }$ after executing an action *a* in the state s.R:S∗A→R, it represents the reward function. For example, R(s,a) represents the reward value given by the environment after the agent executes the action *a* in the state *s*.γ∈[0,1] is a discount factor used to balance the importance of immediate and long-term rewards.

In the remainder of this section the MDP model is formalized based on the MEC environment, and the state space, action space and reward function are subsequently defined. Afterwards, the novel edge server placement algorithm, DQN-ESPA, is introduced.

### 4.1. MDP Model

#### 4.1.1. State

Consider the scenario of placing *m* edge servers in *n* positions, and where only one edge server can be placed in each position. Each placement is a solution to the problem, and it is denoted as a different state. Suppose that the location of each edge server is $\left(lat_{i},lon_{i}\right)$, where $lat_{i}$ is the latitude value of the location of the edge server $s_{i}$, and $lon_{i}$ is the longitude value of the location of the edge server $s_{i}$. The placement sequence will be $\left(\left(lat_{1},lon_{1}\right),\left(lat_{2},lon_{2}\right),\cdots ,\left(lat_{m},lon_{m}\right)\right)$ after edge serves are placed. We define the state space *S*, where $S=\left(\left(lat_{1},lon_{1}\right),\left(lat_{2},lon_{2}\right),\cdots ,\left(lat_{m},lon_{m}\right)\right)$. In particular, the placement sequence varies as the placement of the edge server changes and a new state is generated.

#### 4.1.2. Action

Every time the placement positions of *m* edge servers are randomly selected from *n* positions to get the initial state it is assumed that only one edge server placement is changed at a time. One additional restriction is that each edge server can only move to a nearby location. As an example, consider the diagram shown in Figure 2. There are nine base stations $\left(b_{1},b_{2},\cdots ,b_{9}\right)$, and the edge server $s_{i}$ can be placed at any position of the nine base stations. The initial position of the edge server $s_{i}$ is at the position of the base station $b_{5}$, and the edge server $s_{i}$ can move up, down, left, and right to reach the new placement position. Since the latitude and longitude of each base station are known, the neighbor base stations of each base station are also known. Hence, moving upward means moving in the direction of increasing latitude, moving downward means moving in the direction of decreasing latitude, moving left means moving in the direction of decreasing longitude, and moving right means moving in the direction of increasing longitude.

To normalize the action, the target edge server that needs to be relocated is selected and the direction of its movement is determined. On one side, in the problem of decision-making of the target server, the action space is defined as the set of edge servers. Then the action space can be represented as $A_{1}=\left(0,1,2,\cdots ,n\right)$, where *n* represents the sequence number of the *n*th edge server. On the other side, in the problem of decision-making of the server’s movements, the action space corresponds to the direction set of the server’s movement, which is defined as $A_{2}=\left(0,1,2,3\right)$. 0 means moving upwards, 1 means moving downwards, 2 means left, and 3 means moving right. Therefore, it yields to two discrete action spaces, i.e., $A_{1}$ and $A_{2}$. In order to simplify the problem model, we combine these two action spaces into one, which is the multiplication cross of $A_{1}$ and $A_{2}$. Because there are n edge servers and each edge server can move in 4 directions, there are 4 ∗ *n* actions in total. Therefore, the action space is described as A=(0,1,2,⋯,4∗n). It means that for the input state sequence with *n* edge servers, there are 4∗n output actions, and the optimal action should be selected from 4*n actions.

Based on action *A*, the target edge server that needs to be relocated is calculated as Formula (5).
(5)$$s_{i}=A/4$$

Then the direction of the movement for the edge server $s_{i}$ is calculated as Formula (6), where % is a mod operator.
(6)$$d_{i}=A\%4$$

#### 4.1.3. Reward

We evaluate the placement performance of the edge server from two aspects: average access delay and workload standard deviation. The lower the average delay, the smaller workload standard deviation, indicating the better placement performance. First, the average access delay and the workload standard deviation are standardized. We use the Z-Score standardization method. Assuming a sequence $\left(x_{1},x_{2},\cdots ,x_{n}\right)$, the standardized formula is:(7)$$\left\{\begin{array}{c} \bar{x}=\frac{1}{n}\sum_{i=1}^{n}x_{i}\\ sd=\sqrt{\frac{1}{n-1}\sum_{i=1}^{n}{\left(x_{i}-\bar{x}\right)}^{2}}\\ z_{i}=\frac{x_{i}-\bar{x}}{sd} \end{array}\right.$$

By combining Formulas (2) and (4), the normalization values of the workload balance and the average access delay of the system can be obtained with their respective historical data sets $\left(WSD_{1},WSD_{2},\ldots ,WSD_{n}\right)$ and $\left(MD_{1},MD_{2},\ldots ,MD_{n}\right)$. Then the reward function is shown in Formula (8).
(8)$$R=-(\alpha Z_{WSD}+\beta Z_{MD})$$

Therein, $Z_{WSD}$ and $Z_{MD}$ represent the normalization values of the workload balance and the average access delay, respectively; α and β represent the weights of two factors, and their sum is 1.

### 4.2. DQN-ESPA

Q-learning is a classic non-model-based reinforcement learning algorithm [36] which is used in a variety of problems, including the optimal control problem in the Markov decision process [37], human-machine dialogue [38], robot navigation [39], production scheduling [40], traffic control [41], and so on. The Q-learning algorithm takes the timely reward value *r* leaving the current state and the maximum *Q* value $\max_{a^{\prime }}Q\left(s^{\prime },a^{\prime }\right)$ of the next state $s^{\prime }$ as a complete Markov decision sequence gain. We use the Bellman optimal equation [42] to update the *Q* value, as shown in Formula (9).
(9)$$\begin{array}{c} Q\left(s,a\right)=Q\left(s,a\right)+\alpha \left(s,a\right)(r+\gamma \max_{a^{\prime }}Q\left(s^{\prime },a^{\prime }\right)-Q\left(s,a\right)) \end{array}$$

In this formula, α(s,a)∈[0,1] is the learning rate and it is related to the state and action; γ∈[0,1] is the weight used to balance the timely reward and the long-term reward.

After initializing the edge computing environment, *m* placement positions are randomly selected from the *n* candidate positions, and the initial state is obtained according to the state space defined in the MDP model. At the beginning, the agent executes an action with a random strategy. Each time an action is randomly selected from the action space defined in the MDP model, the agent enters the next state after executing the action. Then the environment gives the agent a timely reward according to the reward function defined in the MDP model. Notice that according to Formula (9) the state action value is updated. This is reflected as the agent continuously interacts with the environment, i.e., the strategies learned by the agent improve persistently, and thus the strategy will converge from a random strategy to the optimal strategy.

During the interaction between the agent and the environment, the agent stores the state action value Q(s,a) of the action *a* executed in the state *s* in a two-dimensional table. When this table has all state action values, the agent can execute the optimal action accordingly. As shown in [16], when the state space is large, a curse of dimensionality problem arises. To overcome this problem we use DNN in our DQN to estimate the value function of Q-learning algorithm [43]. Then, the optimal offload action can be directly obtained by choosing the one that has the maximum *Q*-value. Figure 3 depicts the general DQN flow in our algorithm. Our DQN consists of one hidden layer of 50 neurons.

Our edge server placement algorithm, DQN-ESPA, is described in Algorithm 1. DQN-ESPA updates the θ parameter to let the approximate function $\hat{Q}$ approximately represent the *Q* value, as shown in Formula (10).
(10)$$Q\left(s_{t},a_{t}\right)\approx \hat{Q}\left(s_{t},a_{t},\theta \right)$$

**Algorithm 1** DQN-ESPA
1:Initialization of Replay memory *M*2:Initialization of *Q* function with random parameters θ3:Initialization of $\hat{Q}$ function with parameters $\theta^{-}$4:Randomly select *m* placement positions from *n* candidate placement positions5:Construct the initial state $s_{1}$ based on step 46:

t←1

7:
**while**

t≤T

**do**
8:   generate a random number rnd in [0, 1]9:   **if** rnd>ϵ **then**10:     Select $a_{t}=argmax_{a}Q_{t}\left(s_{t},a;\theta \right)$11:   **else**12:     Select action $a_{t}$ randomly13:   **end if**14:   Execute action $a_{t}$, observe $r_{t}$, $s_{t+1}$15:   Store experience $\left(s_{t},a_{t},r_{t},s_{t+1}\right)$ in *M*16:   Sample minibatch of $\left(s_{t},a_{t},r_{t},s_{t+1}\right)$ randomly from *M*17:   **if**
*t* + 1 is the final one **then**18:     Set $Z_{t}=r_{t}$19:   **else**20:     Set $Z_{t}=r_{t}+\gamma \max_{a_{t+1}}{\hat{Q}}_{t}\left(s_{t+1},a_{t+1};\theta^{-}\right)$21:   **end if**22:   Gradient descent step is executed on ${\left(Z_{t}-Q_{t}\left(s_{t},a;\theta \right)\right)}^{2}$ by θ23:   Reset $\hat{Q}=Q$ in every C steps24:   t←t+125:
**end while**



At time step *t*, the state $s_{t}=\left(\left(lat_{1},lon_{1}\right),\left(lat_{2},lon_{2}\right),\cdots ,\left(lat_{m},lon_{m}\right)\right)$ is set as the input of the neural network to get the current state action $Q_{t}\left(s_{t},a_{t};\theta_{i}\right)$ value, as shown on lines 4 and 5 in Algorithm 1. From line 7 to line 13, the algorithm selects an action according to the ϵ−greedy policy. In order to reduce the connection between the estimated value and the target value and improve training efficiency, DQN-ESPA adds a target network. The target network and the estimated network have the same parameters. Every *C* time steps, the agent assigns the parameters of the estimated network to the target network (line 23 in Algorithm 1). The target value *Z* is defined in Formula (11):(11)$$Z=r_{t}\left(s_{t},a_{t}\right)+\gamma \max_{a_{t}+1}Q_{t}\left(s_{t+1},a_{t+1};\theta_{i}^{-}\right)$$

$\theta_{i}^{-}$ represents the parameters of the target network before a certain iteration time.

The loss function $L_{i}$ in DQN-ESPA is described in Formula (12):(12)$$L_{i}\left(\theta_{i}\right)=E_{s_{t},a_{t},r_{t},s_{t+1}}\left[{\left(Z-Q_{t}\left(s_{t},a_{t};\theta_{t}\right)\right)}^{2}\right]$$

DQN-ESPA uses an experience pool M to store state action transition sequences $\left(s_{t},a_{t},r_{t},s_{t+1}\right)$. According to the experience replay strategy, DQN-ESPA randomly selects a batch of samples from the experience pool for training at each iteration step (lines 15 and 16 in Algorithm 1).

To differentiate the gradient from the loss function, we define it as Formula (13):(13)$$\begin{array}{c} \nabla_{\theta_{i}}L_{i}\left(\theta_{i}\right)=E_{s_{t},a_{t},r_{t},s_{t+1}}\left[\left(Z-Q_{t}\left(s_{t},a_{t};\theta_{t}\right)\right)\nabla_{\theta_{i}}Q_{t}\left(s_{t},a_{t};\theta_{i}\right)\right] \end{array}$$

Thus, the gradient $\nabla_{\theta_{i}}Q_{t}\left(s_{t},a_{t};\theta_{i}\right)$ guides loss function to be reduced at a feasible direction. Finally, we can update the parameters as:(14)$$\theta_{i+1}=\theta_{i+1}+\alpha \nabla_{\theta_{i}}L_{i}\left(\theta_{i}\right)$$
where α is the learning rate in (0, 1).

As it can be seen on line 14 in Algorithm 1, the agent’s experience $\left(s_{t},a_{t},r_{t},s_{t+1}\right)$ is stored in replay memory, therefore there is no need for transition probabilities.

## 5. Performance Evaluation

The average access delay and workload balance are adopted as evaluation indicators to evaluate the performance. The proposed DQN-ESPA is compared to classical algorithms such as SAPA, KMPA, TKPA and RPA. All the experiments were carried out using a real dataset, and the impact of different edge server numbers and placement performance carefully studied.

### 5.1. Configuration of the Experiments

The parameters used in the experiments are set uniformly: the storage space M=100,000, the learning rate α=0.01, the batch size K=64, the parameter update frequency C=100, and the discount factor γ=0.9.

In order to evaluate the overall results of the experiments, the experimental results will be normalized through the log function shown in Formula (15):(15)$$norm_{i}=lg\left(x\right)/lg\left(max\right)$$
where *x* is the sample value that needs to be normalized and max is the maximum value in the sample. Then the overall evaluation indicator is defined as:(16)$$index\left(ad,wb\right)=\mu {norm}_{ad}+\left(1-\mu \right){norm}_{wb}$$

In this formula, ${norm}_{ad}$ is the normalized value of average delay, $norm_{wb}$ is the normalized value of workload balance, and μ is the weighting factor, μ∈[0,1]. In this paper μ is set as μ=0.5.

All the algorithms, including KMPA, TKPA, RPA, and DQN-ESPA, were implemented in Python 3.6, The benchmarks were executed on a computer system with Intel (R)Xeon (R) CPU E5-26200@2.00 Ghz, 128 GB memory and operated in Linux distribution Ubuntu 16.04.

### 5.2. Dataset Description

The dataset used in the experiments was obtained from Shanghai Telecom (http://sguangwang.com/TelecomDataset.html, accessed on 17 February 2022). It contains anonymous calls/Internet information from mobile user requests to 3233 base stations. The exact location of all base stations is included. The dataset contains 4.6 million call records, and 7.5 million Internet traffic records of approximately 10,000 anonymous mobile users gathered for 6 consecutive months. Each call/traffic record contains detailed start time and end time for each mobile user and its corresponding base station. From the Shanghai Telecom dataset we selected the data of 3000 valid base station.

### 5.3. Experimental Results

In order to study the performance effect on the variety of the number of edge servers to be placed, the placement performance are evaluated when the number of edge servers varies between 100 and 300.

The access delay and workload balance of DQN-ESPA are shown in Figure 4a,b when placing 100 edge servers. As the number of iterations increases, the average delay of the entire system continues to decline. It can also be seen that the workload standard deviation continuously decreases. The average access delay and the workload standard deviation of the compared algorithms are shown in Figure 4c,d. The results show that the DQN-ESPA is able to obtain both the lowest average delay and the minimum workload standard deviation.

Figure 5 shows the experimental results when placing 300 edge servers. Therein, Figure 5a,b correspond to the DQN-ESPA delay and workload balance with the variety of the number of iterations. Similar to the previous experiment, the average delay and the workload standard deviation decreases with the rise of the number of iterations. The average delay and the standard deviation of the workload balance of all the compared algorithms are presented in Figure 5c,d. It shows that the DQN-ESPA obtains the lowest average delay. It also shows that KMPA can obtain relative low average delay when placing edge servers, but the workload standard deviation obtained by KMPA is much bigger than the other four algorithms. Meanwhile, TKPA obtains the minimum workload balance standard deviation, but it gets the worse average delay comparing to DQN-ESPA and KMPA.

It can also been seen that DQN-ESPA does not always achieve the best performance on the indicator of workload balance standard deviation, e.g., TKPA and SAPA obtain less workload balance standard deviation placing 300 edge servers in Figure 5d. Therefore, to further compare the algorithms’ performance, we utilize the overall evaluation indicator presented in Formula (16) to evaluate their comprehensive performance, and the results are shown in Table 1. It shows that the comprehensive performance ranking from high to low of the compared algorithms is DQN-ESPA > SAPA > KMPA > TKPA > RPA.

We also introduce the relative comprehensive performance improvement ratio *RP* to evaluate the performance, as shown in Formula (17). Therein, $index_{DQN\_ESPA}$ is the comprehensive performance of DQN-ESPA, while $index_{alg}$ is the comprehensive performance of the compared algorithms, and alg∈{SAPA, KMPA, TKPA, RPA}. When placing 100 edge servers, the overall indicator of DQN-ESPA outperforms SPPA, KMPA, TKPA, and RPA by 1.75%, 5.80%, 12.61%, and 13.40%, respectively. When placing 300 edge servers, the overall indicator of DQN-ESPA outperforms SAPA, KMPA, TKPA, and RPA by 2.39%, 5.22%, 13.26%, and 15.54%, respectively. The results indicate that DQN-ESPA comprehensively considers the average delay and workload balance, and subsequently achieves the best comprehensive performance among the compared algorithms considered in this evaluation.
(17)$$RC=\frac{index_{alg}-index_{DQN\_ESPA}}{index_{alg}}$$

## 6. Conclusions

In this paper we proposed a novel algorithm, DQN-ESPA, for the ESPP in MEC systems. We abstract ESPP in a Markov decision sequence, and the edge server placement problem is solved through deep reinforcement learning. Utilizing a real dataset from Shanghai Telecom, we compared DQN-ESPA to several state-of-the-art algorithms, such as SAPA, KMPA, TKPA and RPA, and use the average access delay and workload balance of the entire system as evaluation indicators. Experimental results show that in comparison to SAPA, KMPA, TKPA and RPA, DQN-ESPA is able to achieve up to 13.40% better performance with 100 edge servers, and up to 15.54% better performance with 300 edge servers.

Although DQN-ESPA surpasses typical comparison algorithms, its convergence speed is relatively slow. This matter will be further studied as future work. In terms of new applications for the proposed algorithm, in the field of edge computing task offloading is another hot topic, and DQN-ESPA can be further studied to solve comprehensive problems for task offloading and the ESPP.

## Figures and Tables

**Figure 1 entropy-24-00317-f001:**
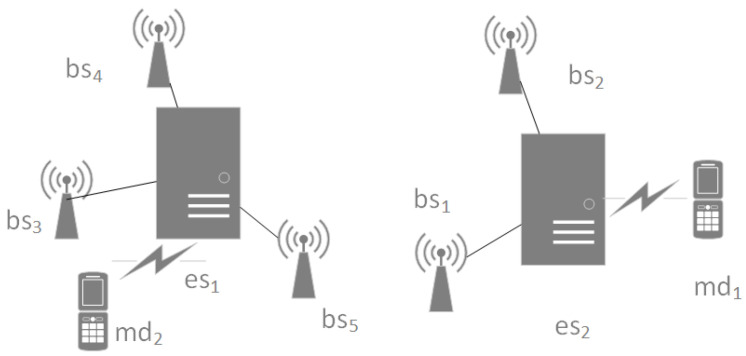
Edge computing system model.

**Figure 2 entropy-24-00317-f002:**
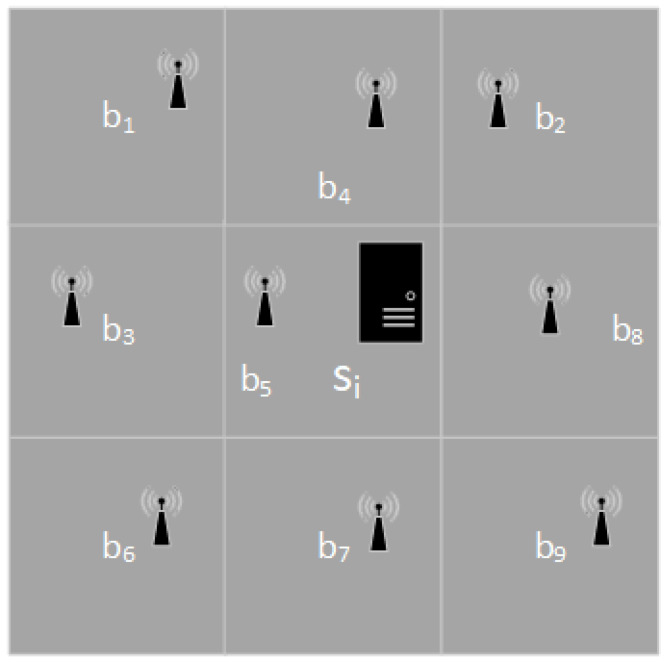
Diagram with location changes of an edge server.

**Figure 3 entropy-24-00317-f003:**
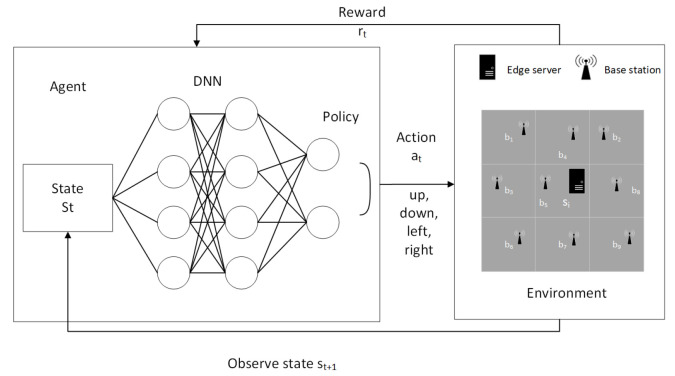
Deep Q-network used in DQN-ESPA.

**Figure 4 entropy-24-00317-f004:**
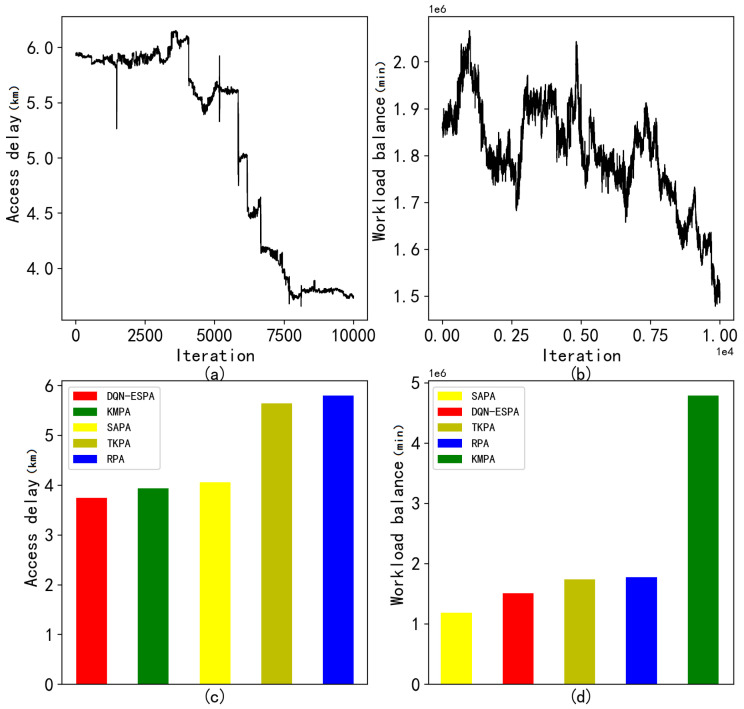
Results with the placement of 100 edge servers at 3000 base station locations. (**a**) Access delay of DQN-ESPA with the variety of iterations; (**b**) Workload balance of DQN-ESPA with the variety of iterations; (**c**) Average access delay of the compared algorithms; (**d**) Average workload standard deviation of the compared algorithms.

**Figure 5 entropy-24-00317-f005:**
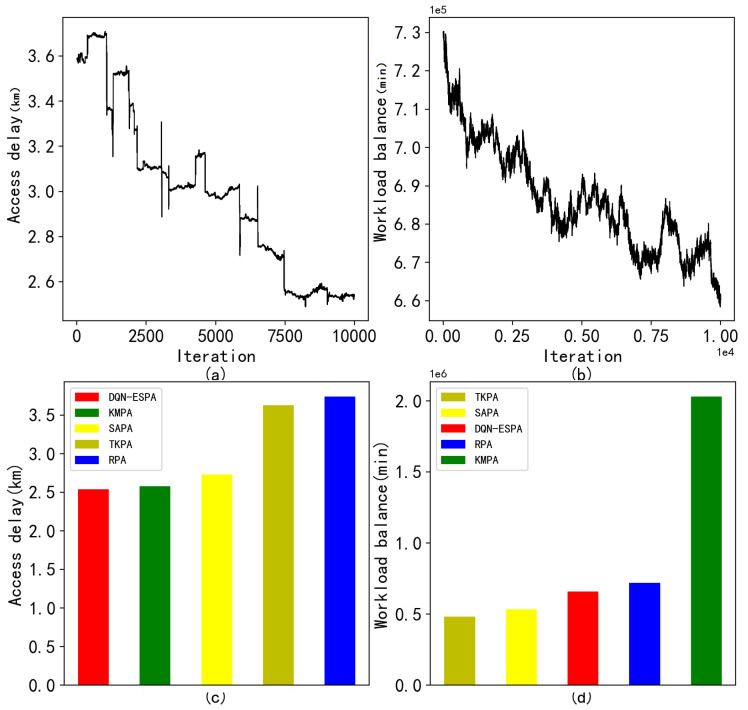
Results with the placement of 300 edge servers at 3000 base station locations. (**a**) Access delay of DQN-ESPA with the variety of iterations; (**b**) Workload balance of DQN-ESPA with the variety of iterations; (**c**) Average access delay of the compared algorithms; (**d**) Average workload standard deviation of the compared algorithms.

**Table 1 entropy-24-00317-t001:** Overall performance indicators.

No.	DQN-ESPA	SAPA	KMPA	TKPA	RPA
100	0.8380	0.8529	0.8896	0.9589	0.9677
300	0.8144	0.8343	0.8593	0.9389	0.9643

## Data Availability

The dataset used in the experiments was obtained from Shanghai Telecom: http://sguangwang.com/TelecomDataset.html, accessed on 17 February 2022.

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
