# Peer review of "An Edge Server Placement Method Based on Reinforcement Learning"

_entropy, 2022, doi:10.3390/e24030317_

Round 1

Reviewer 1 Report

Summary: The paper presents an approach for optimizing edge server placement based on reinforcement learning. Access delay and workload balance are both taken into account for the optimization. Based on real telecom access data, it is shown that the proposed approach provides better results than three state-of-the-art approaches.

The topic of the paper is highly relevant. The paper is overall well written and easy to follow. The description of the system model and the approach appear to be sound – unfortunately, the actual algorithm is not included in the copy of the paper I received, even though it is explained and referenced in the text (as Algorithm 4.2). Without the complete algorithm, it is difficult to make a final assessment on the soundness and, of course, this is a key element that has to be included in the final paper.

The following detailed issues also should be addressed:

  • 1, l.3: “These methods, however, HAVE …”
  • 1, l. 23 – “marginalizing“ does not seem to be the right term (i.e. “to push someone to the side”), I think “localizing” is sufficient here
  • 1, l. 26 “MEC and important partS of the related research HAVE been …”
  • 2, l.70 “ESPA” instead of “ESPP”? – the problem does not select anything, the algorithm does
  • 2, l.78 Reference [22] should be “Wang et al.”?
  • 2, l.83 – the noun “research” does not have a plural in English. If you want an explicit plural, you have to add another noun, e.g. “research activities”, “research projects” or just “approaches”
  • 6, l. 221: “3” should mean “moving right”, not “4”
  • 6, l. 224: The space A is not clearly formulated – what you mean is “m + 4 * n” (m=direction, n=edge server)?
  • 6, Section 4.1.3 – the elements of the reward function (8) need to be clearly described.
  • 8, l.266 reference to Algorithm 1 (instead of Algorithm 4.2 as in other places?), but neither an Algorithm 1 or an Algorithm 4.2 is included in the paper …

Author Response

We would like to express our gratitude to the Entropy journal for a chance to submit our paper, also to the reviewers for the constructive suggestions that helped us improve its quality. A revised version of the paper has been submitted according to the reviewers’ comments and suggestions. Efforts were also made on correcting the mistakes and polishing the language. Detailed replies are given in the attachment.

Reviewer 2 Report

The paper presents a newly developed method for edge server placement in mobile computing systems. The authors use for this purpose a deep neural network with reinforcement learning.

The paper is structured adequately, with introduction, state-of the art survey, methodology description and case study. However, the reviewer's opinion is that the paper must be revised before publication, at least regarding the following issues.

1. The list of references should be revised. For instance, [19] is referred in the text as connected to the Top-K placement algorithm, while its main subject is a mixed-integer linear programming method. The same seems to be true for reference [20], connected by the authors with the random search, having actually as main research approach a genetic algorithm.

2. The authors state that the proposed method overcomes the difficulties associated with MILP, heuristics and metaheuristics ("mixed integer programming algorithms cannot solve large-scale problems in a limited time; heuristic algorithms rely on specific problem models and easily to fall into the local optimal solution; meta-heuristic algorithms require high levels of parameter-tuning"). However, the performance comparison is made against KMPA (k-means), TKPA (top-k placement) and RPA (random placement), which are all simple methods. Moreover, top-k placement can refer to one of several known algorithms. Which was the one used by the authors?

Also, the authors should consider comparing their method to a more complex (MILP or metaheuristic) approach, to make evident the improved performance offered by the new method.

3. At line 280, the authors state that "All the algorithms, including KMPA, TKPA, RPA, and DQN-ESPA, were implemented in Python 3.6, The benchmarks were executed on a computer system with Intel (R)Xeon 290
(R) CPU [email protected] Ghz, 128 GB memory and operated in Linux distribution Ubuntu 16.04."

Such a statement is relevant only in the context of computation time assessment and comparison between methods. This would be a welcomed addition to the paper, since the DQN-ESPA requires a deep neural network training step.

4. At line 328, a table number is missing.

5. At figures 4 and 5, the measurement units of the values represented in the graphics should be provided to make the figures more explanatory.

6. A discussion section should be created or the conclusions should be expanded with an evaluation of the strengths and weaknesses of the proposed method.

Author Response

(The authors gave the same response as above.)

Round 2

Reviewer 1 Report

I have checked the algorithm, which was unreadable in the previous version. I also see that all my minor review comments have been addressed. Thus I now suggest to accept the paper.

Author Response

Thanks for your comments. In addition, the simulated annealing placement algorithm (SAPA)is implemented in the experiments for comparison. Also the results are refreshed in Figure 4 and 5, as well as in Table 1.

Reviewer 2 Report

The reviewer thanks the authors for the changes made in the paper to incorporate the points of concern raised in the first round of review.

Most changes were performed and solved the issues.

But, regarding the algorithms used as comparison, it would be useful if the new method would be tested against existing algorithms with a comparable level of complexity.

Author Response

Thanks for your comments. The simulated annealing placement algorithm (SAPA)is implemented in the experiments for comparison. Also the results are refreshed in Figure 4 and 5, as well as in Table 1.
